# GUIDING EXPLANATORY INFERENCE THROUGH INFERENCE TYPES

## ABSTRACT

This work investigates localised, quasi-symbolic inference behaviours in distributional representation spaces by focusing on Explanation-based Natural Language Inference (NLI), where two explanations (premises) are provided to derive a single conclusion. We first establish the connection between natural language and symbolic inferences by characterising quasi-symbolic NLI behaviours, named *inference types*. Next, we establish the connection between distributional and symbolic inferences by formalising the Transformer NLI model as a rule-based neural NLI model - a *quasi-symbolic NLI framework* where different inference behaviours are encoded as functionally separated subspaces in the latent parametric space. We perform extensive experiments which reveal that inference types can enhance model training and inference dynamics, and deliver localised, symbolic inference control, and latent inference-type disentanglement. Based on these findings, future work will probe the composition and generalisation of symbolic inference behaviour in distributional representation spaces.

## 1 INTRODUCTION

Explanatory sentences (Jansen et al., 2018b; Dalvi et al., 2021) can encode hierarchical, taxonomic, and causal relations between concepts (Gardenfors & Zenker, 2015). By understanding and reasoning over these concepts expressed by explanations, humans can make intricate decisions, which is significant in scientific, cognitive, and AI domains. In this work, we focus on the Explanation-based Natural Language Inference (NLI) task, where two explanations are provided to derive a single conclusion. Within this task, a central challenge involves achieving "localised" and "quasi-symbolic" inference behaviour. E.g., given the two premises: _milk is a kind of liquid_ and _liquids can flow_, one may derive the conclusion _milk can flow_ by localising and substituting the argument _liquids_ with _milk_ or _both liquids and milk can flow_ by conjuncting the arguments.

This localisation behaviour parallels interpretable mechanisms found both in formal inference (e.g., rule-based deductive reasoning) and in the broader field of mechanistic interpretability (Mueller et al., 2025). However, current Transformer-based NLI models generally lack such localisation, as their latent features are often poorly defined and highly entangled, e.g., polysemanticity phenomenon (Scherlis et al., 2022). Moreover, their reasoning behaviour often depends on memorisation rather than generalisation, and rule-based control mechanisms cannot be fully enforced (Yan et al., 2025). To advance the interpretable and controllable NLI, this work investigates a pivotal question: How can "vanilla" Transformer-based NLI models learn and generalise such quasi-symbolic inference in distributional representation spaces? Resolving this question allows us to shorten the gap between formal and material (content-based) inferences (Gildea & Jurafsky, 2000; Banarescu et al., 2013), integrating the flexibility of distributional-neural models with the properties of symbolic, compositional representations, facilitating interpretability, compositionality (Dankers et al., 2022; Marcus, 2003), and reasoning control. Therefore, this work provides a complete initial step in investigating the quasi-symbolic NLI over the distributional latent semantic space.

**First,** we systematically annotate the quasi-symbolic NLI behaviour, grounded on linguistic, formal semantic theory. Valentino et al. (2021) have demonstrated that explanation-based NLI cannot be directly framed as pure logical reasoning, which commonly has subtler incompleteness and consistency problems from a logical point of view. Meanwhile, explanatory NLI corresponding to definable inference patterns and symbolic operations can be localised over the sentence structure. Under

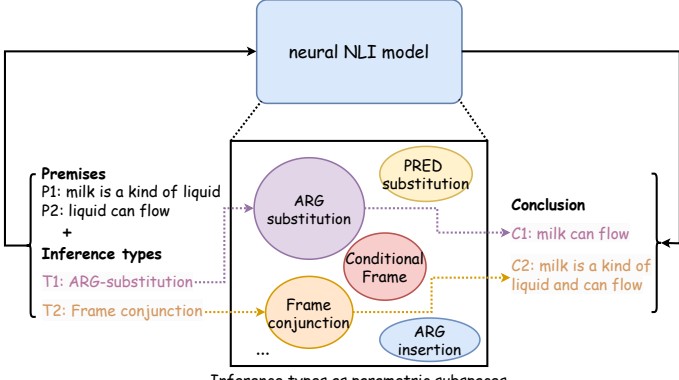

Figure 1: Quasi-symbolic NLI framework. Inference types can be encoded as functional subspaces, which are separated in parametric space. Thus, by manipulating the inference types, we can deliver localised, symbolic inference control.

the Argument Structure Theory (AST) (Jackendoff, 1992), the predicate-argument structure and semantic roles from explanatory sentences can be effectively represented, localised, and disentangled in the latent space of transformer-based models (Zhang et al., 2024a;c). A particular instance of an AST representation is the Abstract Meaning Representation (AMR) (Banarescu et al., 2013).

Motivated by this middle ground between logical representations and lexico-semantic inference patterns, we introduce granular inference types based on explanatory sentences, using AMR as an interface between formal and material inferences to define the symbolic operations. Specifically, *we leverage the AMR to systematically characterise quasi-symbolic inference behaviours, named inference types, grounded on AMR symbolic graphs.* Using the explanation-based NLI dataset (EntailmentBank), we identify ten categories of symbolic transformations and provide annotations for 5,134 premise-conclusion pairs. In this initial study, we exclude AMR representations from the input to minimise the influence of explicit structural linguistic biases on the internal geometry learned by the vanilla Transformer NLI model. Instead, only the inference-type labels are provided as input.

**Second,** to enable localised, quasi-symbolic NLI control, a vanilla neural NLI model, such as T5 (Raffel et al., 2020), should function as a soft rule-based system in which inference types are explicitly encoded and separated within the latent space. Then, providing the same input premises with different inference type specifications should yield conclusions that preserve the same lexical semantics but differ in sentence-level semantics corresponding to different inference types, reflecting the localised, quasi-symbolic inference control. Motivated by this, *we propose a quasi-symbolic NLI framework under Neural Tangent Kernel (NTK) theory* (Jacot et al., 2018) where different functions (input-output inference type transformation) are encoded as separated subspaces in the parametric space via gradient descent optimisation. NTK theory offers a principled interpretation of training dynamics, knowledge memorisation, retrieval, and representation, showing that generalisation and inference behaviours are shaped by the geometric structure of gradients throughout training (Ortiz-Jimenez et al., 2023). In this work, we adopt NTK as a conceptual lens, rather than a practical implementation, to more effectively interpret the behaviour of the vanilla Transformer NLI model.

To assess this conceptual framework, we prefix premises with inference-type labels to condition model behaviour. After training, the model should exhibit the behaviours, including: (1) Training Dynamics: During training, explicit supervision on inference types aligns the model's reasoning trajectory with target inference behaviours, improving conclusion prediction accuracy. (2) Inference Composition: By varying inference type during inference, the model can separate the semantics of the premises from the inference behaviour. This enables localised, quasi-symbolic NLI control, allowing for flexible and interpretable reasoning. (3) Inference-type Separation: The inference-type can be clustered and separated within the latent space. This finding provides a more nuanced understanding from previous work (Kumar et al., 2025), suggesting that higher task performance does not necessarily correspond to more structured or disentangled internal representations. Instead, the learned function space may become increasingly entangled as a result of SGD optimisation.

In summary, this work provides a foundation in investigating the quasi-symbolic NLI over distributional semantic spaces, with the following contributions:

**Linguistic Formalisation:** Systematic characterisation of inference types via AMR, bridging formal and material inference. The annotations are planned for public release.

**Theoretical Framework:** A quasi-symbolic NLI framework based on NTK theory where inference types govern subspace formation in latent representations.

**Empirical Validation:** Demonstrated improvements in training efficiency, inference accuracy, and localised control, suggesting future direction of rule-based learning and generalising in neural spaces. The experimental pipelines are released `https://anonymous.4open.science/r/Inference_type-5E07/`.

## 2 RELATED WORK

In this section, we review the related work around two topics: *neuro-symbolic representations* and *semantic control over latent spaces*, to highlight the current research limitation and elucidate the motivation underlying our work.

**Neuro-symbolic representations.** A longstanding goal in the neuro-symbolic domain is to blend the representational strengths of neural networks with the interpretability of symbolic systems to build more robust NLI models. Current methods usually inject symbolic behaviour through explicit symbolic representations, including graph (Khashabi et al., 2018; Khot et al., 2017; Jansen et al., 2017; Kalouli et al., 2020; Thayaparan et al., 2021), linear programming (Valentino et al., 2022b; Thayaparan et al., 2024), adopting iterative methods, using sparse encoding mechanisms (Valentino et al., 2020; Lin et al., 2020), synthetic quasi-natural language expression (Clark et al., 2020; Yang & Deng, 2021; Yanaka et al., 2021; Fu & Frank, 2024; Weir et al., 2024), symbolic-refined LLMs (Olausson et al., 2023; Quan et al., 2024), etc. Those studies ignore the underlying neuro-symbolic behaviour in the representation space. From an Explainable AI perspective, many studies have shown that neural networks can encode sparse neuro-symbolic concepts without explicit symbolic injection across areas like image embedding (Ren et al., 2022; Deng et al., 2021; Li & Zhang, 2023), word embedding (Ethayarajh et al., 2018; Allen et al., 2019; Ri et al., 2023), and contextual embedding (Gurnee et al., 2023; Nanda et al., 2023; Li et al., 2024; Park et al., 2024; Templeton et al., 2024). By understanding the symbolic behaviour within neural networks, their decision-making logic can be better interpreted and controlled (Chen et al., 2024).

In this work, we draw on quasi-symbolic NLI objectives within distributional neural models, targeting better reasoning controllability and interpretability.

**Semantic control over latent spaces.** Latent variable models, such as VAE (Kingma & Welling, 2013) and Diffusion (Dhariwal & Nichol, 2021), have shown the capability of symbolic representation, control, and interpretation over the distributional space, which are widely deployed in the NLP domain, such as disentangled representation learning (Zhang et al., 2024a) and style-transfer (Gu et al., 2023; Zhang et al., 2024b). Guided by semantic annotation, such as labels (Carvalho et al., 2023) and classifiers (Ho & Salimans, 2022), distinct semantic features can be geometrically separated and composed in the latent space, enhancing localisation and interpretability. However, this concept remains under-explored in the NLI domain. Thus, we propose the quasi-symbolic NLI framework and inference types as an initial step to probe the localised, quasi-symbolic NLI behaviour within vanilla neural NLI models.

## 3 DEFINING INFERENCE TYPES

In this section, we introduce a set of granular inference types derived from explanatory sentences, using AMR to define the symbolic operations. These operations capture the transformations from premises to conclusions at a semantic level. It is important to note that AMR is not employed as a representational component within the proposed model architecture. Rather, it serves as a formal semantic framework to precisely ground and characterise the symbolic inference operations. Table 1 presents the AMR-grounded inference types alongside illustrative examples from the En-

Table 1: Examples of symbolic inference types, with their corresponding abbreviations provided in parentheses and used consistently throughout the paper. The EntailmentBank utilised for this task comprises 5,134 (premises, conclusion) instances with our annotations. These annotations are planned for public release.

| Original Type | Symbolic Type | Prop. | Example Entailment Relation |
|---|---|---|---|
| Substitution | ARG substitution (ARG-SUB) | 19% | P1: a scar on the knee is a kind of scar
P2: a scar is an acquired characteristic
C: a scar on the knee is an acquired characteristic |
| | PRED substitution (PRED-SUB) | 5% | P1: food contains nutrients and energy for living things
P2: to contain something can mean to store something
C: food stores nutrients and energy for living things |
| | Frame substitution (FRAME-SUB) | 20% | P1: the formation of diamonds requires intense pressure
P2: the pressure is intense deep below earth 's crust
C: the formation of diamonds occurs deep below the crust of the earth |
| Further Specification or Conjunction | ARG insertion (ARG-INS) | 18% | P1: solar energy comes from the sun
P2: solar energy is a kind of energy
P3: solar energy is a kind of energy that comes from the sun |
| | Frame conjunction (FRAME-CONJ) | 6% | P1: photosynthesis stores energy
P2: respiration releases energy
C: photosynthesis stores energy and respiration releases energy |
| Inference from Rule | Conditional frame insertion/substitution (COND-FRAME) | 12% | P1: if something is renewable then that something is not a fossil
P2: fuel wood is a renewable resource
C: wood is not a fossil fuel |
| Infer Class from Properties | ARG/PRED generalisation (ARG/PRED-GEN) | 1% | P1: rock is a hard material
P2: granite is a hard material
C: granite is a kind of rock |
| Property Inheritance | ARG substitution (Property Inheritance) (ARG-SUB-PROP) | 0.4% | P1: blacktop is made of asphalt concrete
P2: asphalt has a smooth surface
C: a blacktop has a smooth surface |
| Causal Expression | Causality (IFT) | 0.8% | an optical telescope requires visible light for human to use
clouds / dusts block visible light
if there is clouds or dusts, then the optical telescope cannot be used |
| Example-based Inference | Example (EXAMPLE) | 0.9% | a shelter can be used for living in by raccoons
some raccoons live in hollow logs
an example of a shelter is a raccoon living in a hollow log |

tailmentBank corpus. In what follows, we formally define each lexico-semantic inference type and its corresponding symbolic transformation.

The *substitution* category refers to obtaining a conclusion by replacing a predicate/argument term from one premise with a predicate/argument term from the other premise. Possible variations of this category include (1) *argument (ARG) substitution*, (2) *predicate (PRED) substitution*, and (3) *frame (PRED+ARG) substitution*. In this category, one premise is used to connect two terms which are usually connected by *is a kind of, is a source of*, etc. Conceptualising the AMR representation as a graph, this can be symbolically represented as a subgraph substitution operation over the premise graphs, as illustrated in Figure 2. The *PRED substitution* category works in a similar manner, but replacing a predicate term. The two predicates are usually linked by the following patterns: "$v_1$ *is a kind of* $v_2$", "*to* $v_1$ *something means to* $v_2$ *something*", etc. The *frame (PRED+ARG) substitution* category combines both previous categories by replacing a frame (predicate subgraph) of one of the premises with one from the other premise.

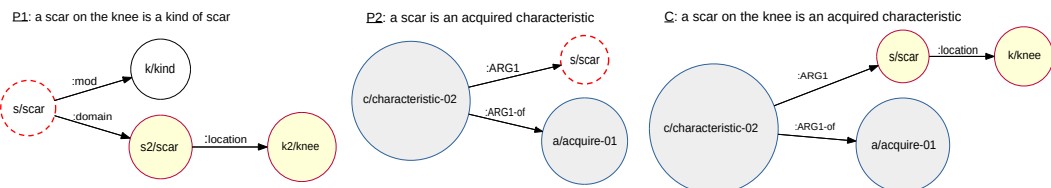

Figure 2: AMR argument substitution: the inference behaviour is defined as subgraph substitution.

The *further specification or conjunction* category allows for obtaining a conclusion by joining both premises. It includes (4) *ARG insertion* and (5) *frame conjunction*. For *ARG insertion*, the conclu-

sion is obtained by connecting an argument from one of the premises to a frame of the other. As for *frame conjunction/disjunction*, the conclusion is obtained by joining the premises graphs through a conjunction/disjunction node (*and*) or (*or*).

The *inference from rule* category from Dalvi et al. (2021) encompasses a specific instance of insertion or substitution, identified as (6) *conditional frame insertion/substitution*. In this category, a frame is either inserted or replaced as an argument of a premise, following a conditional pathway present in the other premise.

The inference type *infer class from properties* has been re-categorised as (7) *ARG or PRED generalisation*, where a new *:domain* relation frame is created if both premise graphs differ by a single predicate/argument term. (8) *Property inheritance*, on the other hand, is a special case of *ARG substitution*, where one of the premises describes a *is made of* relationship between the entity in the other premise and its replacement.

Finally, (9) *Causal Expression* and (10) *Example-based Inference* categories are defined according to the key lexical characteristic of the conclusion, as systematic AMR transformations which could be applied without rephrasing the underlying explanatory sentences could not be determined. More details about the annotation procedure are provided in the supplementary materials.

Thus far, we have defined the explanatory NLI behaviours grounded on the symbolic AMR graph from the perspective of formal semantic theory. In the next section, we aim to introduce the quasi-symbolic NLI framework.

## 4 QUASI-SYMBOLIC NLI FRAMEWORK

### 4.1 QUASI-SYMBOLIC NLI FORMALISATION

In this study, we formalise the "quasi-symbolic NLI behaviour" as rule-based reasoning over neural representation, where discrete inference behaviours are implemented through differentiable transformations over continuous neural representations. This is achieved by characterising and manipulating quasi-symbolic inference behaviours, denoted by $\pi \in \Pi$, where $\Pi$ represents the space of all possible inference rules. The process involves three key stages: (i) Neural Encoding: The premises $p_1$ and $p_2$ are encoded into continuous vector representations: $\overrightarrow{p_1}, \overrightarrow{p_2} = f_{encode}(p_1, p_2)$ (ii) Rule-Based Reasoning: The encoded representations are transformed using a reasoning function guided by the inference behaviour $\pi$: $\overrightarrow{c} = f_{reason}(\overrightarrow{p_1}, \overrightarrow{p_2}; \pi)$ (iii) Neural Decoding: The resulting vector, $\overrightarrow{c}$, is decoded into a natural language conclusion $c$: $c = f_{decode}(\overrightarrow{c})$. Here, $f_{encode}, f_{reason}, f_{decode}$ represent the encoding, reasoning, and decoding functions in a neural NLI model. The injection of $\pi$ should exhibit three advantages:

*1. Training Dynamics:* During training, explicit supervision on $\pi$ aligns the model's reasoning trajectory with target inference behaviours, improving prediction accuracy.

*2. Inference Composition:* By varying $\pi$ during inference, the model can separate the semantics of the premises from the inference behaviour. This enables localised, quasi-symbolic NLI control.

*3. Inference-type Separation:* Explicitly injecting $\pi$ leads to inference-type separation within the latent space, improving geometrical interpretability.

### 4.2 QUASI-SYMBOLIC NLI REPRESENTATION

We focus on standard neural NLI models, with particular, but not exclusive, attention to encoder-decoder architectures such as T5, due to their inherent separation of reasoning and decoding phases, which naturally accommodates quasi-symbolic reasoning. However, this framework can also be adapted into the decoder-only architecture, where the rules are captured, such as through in-context learning (Liu et al., 2024). From a representational perspective, we propose the concepts of latent rule space and feature space to align with the function of the neuro-symbolic NLI model.

**Latent rule space.** The latent rule space refers to the functional parametric space (i.e., models' weights), which captures the structured, rule-based reasoning behaviours $\pi \in \Pi$.

***Proposition1:*** *The inference-types can be encoded and separated within the parametric space.*

We further propose that rule-based reasoning is primarily materialised in the encoder.

***Proposition2:*** *The inference behaviour is instantiated at the encoder and can be controlled by the injection of the associated inference type labels.*

**Latent feature space.**    The latent feature space refers to the output embedding space. To evaluate the feature representation capability, we next describe the methodological framework behind the construction of the abstract sentence representation within T5 (named T5 bottleneck). As for the encoder's final layer output embedding, we compute dimension-wise mean pooling over token embeddings, followed by a multi-layer perceptron to obtain sentence embeddings. As for the decoder's input embedding, we reconstruct token embeddings via linear projection, feeding them into the decoder's cross-attention mechanism. Here, we only describe the optimal setup. We provide a systematic way to choose the best setup in the Appendix.

### 4.3   PROPOSITION1: FORMAL PROOF AND EVALUATION

**NTK interpretation of inference-type subspaces.**    Each symbolic inference type $\pi$ is explicitly embedded as part of the model input, for example, as a token prefix (EP). As a result, the model effectively learns a function $f_\theta(x, \pi)$, where $x = (p_1, p_2)$ are the premises and $\pi$ is the symbolic inference type. The function $f_\theta$ thus jointly depends on both the content of the premises and the nature of the symbolic operation to be performed.

Within the Neural Tangent Kernel (NTK) framework, the similarity between two input examples of the same inference type $\pi$ is captured by the NTK as follows:

$$\Theta_\pi(x, x') = \nabla_\theta f_\theta(x, \pi)^\top \nabla_\theta f_\theta(x', \pi) \tag{1}$$

where $\nabla_\theta f_\theta(x, \pi)$ denotes the gradient of the model output with respect to its parameters, evaluated at the input $(x, \pi)$. This kernel quantifies how a parameter update from one input-output pair would affect another pair, conditioned on the shared inference type.

According to NTK theory (Jacot et al., 2018), in the infinite-width limit, the evolution of the model's predictions under gradient descent training can be described by a linear kernel regression in the RKHS (Reproducing Kernel Hilbert Space) associated with $\Theta_\pi$. Specifically, the prediction at time $t$, $f_t(x, \pi)$, evolves as: $f_t(x, \pi) = f_0(x, \pi) - \Theta_\pi(x, \cdot) [\Theta_\pi + \lambda I]^{-1} (f_0 - c)$ where $f_0(x, \pi)$ is the model's output at initialisation for each training input, $\lambda$ is a regularisation parameter, and $c$ is the vector of ground truth conclusions.

Crucially, this formulation implies that each symbolic inference type $\pi$ induces a distinct kernel $\Theta_\pi$, which in turn defines a unique RKHS $\mathcal{H}_\pi$, that is, a function space within which the model's solutions for inference-type $\pi$ reside. As the symbolic type $\pi$ is varied, the structure of the kernel and the corresponding function space changes, reflecting the distinct reasoning behaviours or transformations associated with different inference operations. Thus, the model encodes different symbolic inference patterns in distinct, kernel-induced subspaces.

**Quantitative evaluation.**    For two different inference types, $\pi_i \neq \pi_j$, we examine the relationship between their corresponding neural tangent kernels (NTKs), $\Theta_{\pi_i}$ and $\Theta_{\pi_j}$. Specifically, we are interested in the interaction between the parameter gradients induced by inputs associated with different inference types.

Consider two data points $x$ and $x'$, possibly corresponding to different premise pairs. When considering cross-type similarities, we are interested in the inner product between the gradients for different types:

$$G_{ij}(x, x') := \langle \nabla_\theta f_\theta(x, \pi_i), \nabla_\theta f_\theta(x', \pi_j) \rangle \tag{2}$$

If the symbolic inference types $\pi_i$ and $\pi_j$ encode fundamentally different reasoning operations (e.g., ARG-SUB vs. PRED-SUB), the gradients with respect to $\theta$ for inputs labelled with $\pi_i$ and those labelled with $\pi_j$ will tend to point in different directions (i.e., orthogonality) in parameter space. This is because each type imposes a distinct task or transformation pattern on the model, causing it to utilise different portions of its capacity. Therefore, by measuring the cosine similarity between gradient vectors associated with different inference types, we can quantify the separability between different inference-type subspaces, as provided in Figure 3.

## 5 EMPIRICAL ANALYSIS

The experiment addresses four key questions: (i) Do symbolic inference types enhance model training and inference performance? (ii) Can these inference types be used for prescriptive inference control? (iii) Does the incorporation of a sentence bottleneck contribute to improved feature representation? (iv) Whether the inference-type can be separated and clustered in the latent space? Due to the page limitation, all experimental details and additional results are provided in the Appendix.

### 5.1 TRAINING AND INFERENCE EVALUATION

First, we evaluate (i) if symbolic inference types enhance model training and inference performance. We consider three mechanisms to inject the inference types into the model. **i.** The inference type as the prefix for the premises at the Encoder; **ii.** The inference type as the prefix for the conclusion in the Decoder; **iii.** The inference type at the end of the conclusion in the Decoder.

**Training dynamics.** We first evaluate generative performance after training based on three metrics: BLEURT (Sellam et al., 2020), BLEU (Papineni et al., 2002), and cosine similarity against sentenceT5 (Ni et al., 2021). By comparing the predicted and golden conclusions, we can fairly evaluate the accuracy of the NLI performance. For the baseline, we choose the T5, GPT2 (Radford et al., 2019), Qwen2.5 (Qwen et al., 2025), Llama3.2 (Grattafiori et al., 2024), our T5 bottleneck and Optimus (Li et al., 2020) with 768 latent dimensions as testbed. The performances are measured from the Entailment test set.

Table 2: Quantitative evaluation on testset, where best results are highlighted in **bold**. Specification for abbreviation. INJ: ways for injecting the information of inference types into the model, including DE: decoder end, DP: decoder prefix, EP: encoder prefix, NO: no inference type. We can observe that injecting inference-type can help model training.

| Baseline | INJ | BLEU | Cosine | BLEURT |
|---|---|---|---|---|
| _encoder-decoder architecture_ | | | | |
| T5 original (small) | DE | 0.55 | 0.96 | 0.30 |
| | DP | 0.59 | 0.96 | 0.34 |
| | EP | **0.65** | **0.97** | **0.45** |
| | NO | 0.54 | 0.96 | 0.22 |
| T5 original (large) | DE | 0.60 | 0.97 | 0.46 |
| | DP | 0.64 | 0.97 | 0.44 |
| | EP | **0.67** | **0.97** | **0.50** |
| | NO | 0.57 | 0.96 | 0.31 |
| _decoder only architecture_ | | | | |
| GPT2(xl) | DP | **0.28** | **0.91** | **-0.90** |
| | NO | 0.27 | 0.90 | -0.97 |
| Qwen2.5(0.5B) | DP | **0.65** | **0.97** | **0.48** |
| | NO | 0.63 | 0.97 | 0.45 |
| Llama3.2(1B) | DP | **0.63** | **0.97** | **0.46** |
| | NO | 0.60 | 0.96 | 0.42 |
| _encoder-bottleneck-decoder_ | | | | |
| T5 bottleneck (base) | DE | 0.35 | 0.91 | -0.15 |
| | DP | 0.39 | 0.91 | -0.13 |
| | EP | **0.42** | **0.92** | **-0.07** |
| | NO | 0.35 | 0.91 | -0.20 |
| Optimus (BERT-GPT2) | DE | **0.26** | **0.80** | **-1.11** |
| | DP | 0.25 | 0.79 | -1.14 |
| | EP | 0.09 | 0.74 | -1.17 |
| | NO | 0.07 | 0.74 | -1.20 |

Table 3: Agreement scores for the quantitative analysis using LLMs on the test set. We also provide a qualitative manual evaluation in the Appendix.

| Baseline | INJ | ChatGPT4o | GPT4o-mini |
|---|---|---|---|
| T5 original (large) | DE | 0.85 | 0.83 |
| | DP | 0.86 | 0.83 |
| | EP | **0.91** | **0.85** |
| | NO | 0.84 | 0.82 |

Table 4: ICL evaluation of test cases, where worst results are highlighted in underline. The prompt is "performing natural language inference [where the inference type is type, description], $[p1; p2; c]_{\times N}$. p1, p2, what is the conclusion?". $N$ is the number of examples. The description is based on the definition of inference types in Section 3.

| Baseline | INJ | N | BLEU | Cosine | BLEURT |
|---|---|---|---|---|---|
| _encoder-decoder architecture_ | | | | | |
| CoT-T5 (11b) (Kim et al., 2023) | Yes | 10 | 0.51 | 0.97 | 0.39 |
| | Yes | 5 | 0.51 | 0.97 | 0.39 |
| | Yes | 0 | 0.50 | 0.97 | 0.36 |
| | NO | 0 | 0.46 | 0.96 | 0.31 |
| Flan-T5 (xxl) | Yes | 10 | 0.51 | 0.97 | 0.41 |
| | Yes | 5 | 0.53 | 0.97 | 0.43 |
| | Yes | 0 | 0.50 | 0.96 | 0.37 |
| | NO | 0 | 0.48 | 0.96 | 0.36 |
| _decoder only architecture_ | | | | | |
| GPT-4-0613 | Yes | 10 | 0.53 | 0.97 | 0.50 |
| | Yes | 5 | 0.52 | 0.97 | 0.47 |
| | Yes | 0 | 0.52 | 0.97 | 0.50 |
| | NO | 0 | 0.47 | 0.96 | 0.40 |
| llama3-70b-8192 | Yes | 10 | 0.54 | 0.97 | 0.54 |
| | Yes | 5 | 0.52 | 0.97 | 0.52 |
| | Yes | 0 | 0.51 | 0.97 | 0.47 |
| | NO | 0 | 0.44 | 0.96 | 0.40 |

In Table 2, across all baseline models, incorporating inference types into the encoder consistently results in improved performance as measured by BLEU, Cosine, and BLEURT metrics, indicating the inference behaviour is instantiated at the encoder (*Proposition*) (**Finding 1**). This finding also suggests that during training, explicit supervision on inference types aligns the model's reasoning trajectory with target inference behaviours, improving conclusion prediction accuracy (**Finding 2**). A similar observation is reflected in the test loss curve shown in Figure 9 in the Appendix.

Furthermore, previous studies have revealed that the LLM evaluation can be consistent with the results obtained by expert human evaluation (Chiang & Lee, 2023; Liu et al., 2023; Wang et al., 2023; Huang et al., 2023). Thus, we also conduct a quantitative analysis to measure whether the generated conclusion contradicts the premises through LLM evaluators, including ChatGPT4o as the baseline and GPT4o-mini for comparison. Table 3 indicates that EP can consistently result in improved LLM agreement scores. A manual check is presented in the Appendix (Tables 13 and 14).

**In-context learning.** Next, we quantitatively evaluate the inference types within in-context learning (ICL) in contemporary large language models (LLMs). As illustrated in Table 4, prompting with inference types can improve the performance of ICL in all tested LLMs. Besides, within the context of causal LLMs, an increase in few-shot examples[1], improves the performance. This finding indicates that our inference types can be generalised across various checkpoints and architectures, ultimately enhancing the reasoning capabilities of LLMs (**Finding 3**).

## 5.2 QUASI-SYMBOLIC NLI EVALUATION

Second, we evaluate (ii) if these inference types can be used for prescriptive inference control.

**Qualitative evaluation.** We qualitatively evaluate the quasi-symbolic NLI behaviour on the generation of conclusions by systematically intervening on the inference type prior to the encoder. As illustrated in Table 5, we can observe that the associated linguistic properties of the conclusion can be controlled consistently with the inference type modifications and this inference control is independent of the semantics of premises, which indicates that the representation mechanisms can improve inference control with regard to symbolic/lexico-semantic properties (**Finding 4**). For example, when the type is ARG substitution (ARG-SUB), the model can generate *the blacktop is made of a smooth surface* by replacing the argument *asphalt concrete* with *smooth surface*. The conclusions are changed to *asphalt and blacktop have the same surface* when the inference type is the conjunction (FRAME-CONJ). Additional examples are provided in Table 15.

**Quantitative analysis.** Next, we perform an automated quantitative analysis using LLMs, including ChatGPT4o and GPT4o-mini. For each pair of premises in the EntailmentBank test set, we apply various inference types to generate a diverse set of conclusions using the fine-tuned T5 (base) model. We then assess the resulting (premises, conclusion, inference type) tuples based on two criteria: (i) whether the generated conclusion contradicts the premises, and (ii) whether the (premises, conclusion) pair is consistent with the specified inference type. Utilising the prompt detailed in Table 16, we report the model agreement score for each criterion. As illustrated in Table 6, the T5 (base) model with controlled symbolic inference types achieves consistency and alignment scores exceeding 60% for both evaluated dimensions.

## 5.3 LATENT FEATURE SPACE EVALUATION

Third, we evaluate (iii) whether the incorporation of feature space (i.e., abstract sentence bottleneck) contributes to improved feature, concept representation in the NLI task. We especially select the VAE baselines due to their analogous encoder-bottleneck-decoder architecture, wherein the compressed sentence bottleneck captures high-level, generalised semantics (concepts) (Mercatali & Freitas, 2021; Zhang et al., 2024a) and evaluate the abstract sentence embedding using as an associated explanation retrieval task (explanation-regeneration - i.e. retrieving the associated explanatory facts relevant to a claim) (Valentino et al., 2022a). Given a question-and-answer pair, it reconstructs the entailment tree by searching the explanations from a fact bank (i.e., WorldTree (Jansen et al.,

---

[1]We randomly sample the examples with the same inference type as the current test example from the training set. We perform ten times and calculate the average for each metric.

Table 5: Qualitative evaluation over original T5 (base) (ARG-SUB: argument substitution, ARG/PRED-GEN: argument/predicate generalisation. ARG-SUB-PROP: property inheritance. ARG-INS: argument insertion, FRAME-CON: frame conjunction, IFT: casual expression).

> P1: blacktop is made of asphalt concrete
> P2: asphalt has a smooth surface
>
> ARG-SUB: the blacktop is made of smooth surface
> ARG-SUB-PROP: blacktop has a smooth surface
> ARG/PRED-GEN: a blacktop is a kind of asphalt
> ARG-INS: asphalt concrete blacktop has a smooth surface
> FRAME-CON: asphalt and blacktop have the same surface
> IFT: if the asphalt has a smooth surface then the blacktop will have a smooth surface

Table 6: Quantitative evaluation, where (i) consistency: whether the generated conclusion contradicts the premises, (ii) alignment: whether the (premises, conclusion) pair is consistent with the specified inference type.

| Evaluators | consistency | alignment |
|---|---|---|
| ChatGPT4o | 0.67 | 0.63 |
| GPT4o-mini | 0.65 | 0.62 |

Table 7: Explanatory inference retrieval task where t represents the depth of entailment tree.

| depth | t=1 | t=2 | t=3 | t=4 |
|---|---|---|---|---|
| DAE | 30.27 | 31.74 | 30.65 | 30.74 |
| AAE | 29.13 | 30.47 | 29.33 | 29.14 |
| DAAE | 13.16 | 15.42 | 14.30 | 13.97 |
| $\beta$-VAE | 10.03 | 10.07 | 10.05 | 10.05 |
| Optimus | 28.21 | 29.35 | 28.35 | 28.27 |
| T5 bottleneck | **34.47** | **35.28** | **34.50** | **34.47** |

2018a)) in an iterative fashion using a dense sentence encoder. In this framework, we can replace the sentence embeddings with the proposed T5 bottleneck baseline to evaluate its abstract sentence embeddings. We compare the T5 bottleneck with abstract sentence representation baselines: Optimus and four LSTM VAEs, and evaluate them via mean average precision (MAP). As illustrated in Table 7, the T5 bottleneck outperforms all baselines, indicating that it can deliver a better abstract representation of explanatory sentences and entailment relations in a retrieval setting **(Finding 5)**.

### 5.4 LATENT INFERENCE-TYPE SEPARATION

Finally, we evaluate the separability of inference-types in the latent space. For the latent parametric space, we measure the cosine similarity between gradient vectors associated with different inference types. As shown in Figure 3 (left), we can observe that when injecting inference-type categories into the model during training, the diagonal values exhibit higher values, indicating the inference-type subspaces can be better separated in the parameter space **(Finding 6)**.

Next, we evaluate whether inference rules exhibit separability within the latent sentence space. We jointly train the latent space with a linear classifier to predict the inference type categories. As shown in Figure 3 (right), our results indicate that inference types can be partially clustered and separated within this latent space, suggesting the feasibility of rule-based learning through appropriate optimisation strategies (Xie et al., 2025) or architectural designs, such as disentangling rules from lexical semantics **(Finding 7)**.

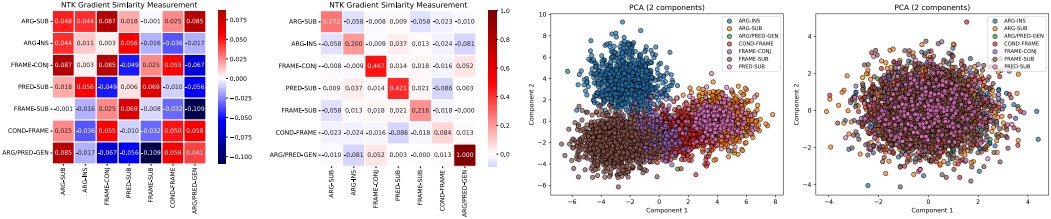

Figure 3: Latent space visualisation. Left: quantitative measuring the separability between different inference-type subspaces in T5 (small) where left: NO, right: encoder prefix (EP); Right: PCA visualisation: inference types cluster and separation, where left: EP, right: NO.

## 6 CONCLUSION AND FUTURE WORKS

This study serves as a foundational step in exploring the quasi-symbolic NLI behaviour within distributional latent semantic spaces. We establish the connection between natural and symbolic language inferences by characterising quasi-symbolic inference behaviours based on AMR graphs. Then, we propose the quasi-symbolic NLI framework based on the NTK theory, where distinct inference types can be represented as separated functional subspaces within the parametric space. Experimental results reveal that integrating symbolic inference types enhances training dynamics, inference precision, explanation retrieval, and inference-type disentanglement, suggesting the potential for neuro-symbolic NLI. In future work, we will further investigate this hypothesis across a broader range of reasoning tasks, such as mathematical reasoning (Meadows et al., 2024), and by incorporating structural AMR representations as inputs via a graph encoder. By more tightly integrating formal linguistic structures with neural representations, we aim to develop a more explainable and controllable neuro-symbolic NLI model.

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

## A  ANNOTATION DETAILS

**Annotation procedure.**  Annotation was performed manually for 5134 entailment triples (two premises, one conclusion) from the EntailmentBank (Dalvi et al., 2021), according to Algorithm 1. Graph subset relations and root matching are relaxed for non-argument (:ARG*, op*) edges, meaning relations such as *:manner* or *:time* can be ignored for this purpose. Two independent annotators with post-graduate level backgrounds in Computational Linguistics were used in this process, on a consensus-based annotation scheme where a first annotator defined the transformations and a second annotator verified and refined the annotation scheme, in two iterations. The annotation of the AMR graph is based on an off-the-shelf parser (Damonte et al., 2017). The descriptions for each inference type category are as follows:

**ARG-SUB** (Figure 2): the conclusion is obtained by replacing one argument with another argument. **PRED-SUB**: the conclusion is obtained by replacing one verb with another verb. **FRAME-SUB**: the conclusion is obtained by replacing a frame of one of the premises with one from the other premise.

**COND-FRAM** (Figure 4): the conclusion is obtained according to the conditional premise with keyword "if".

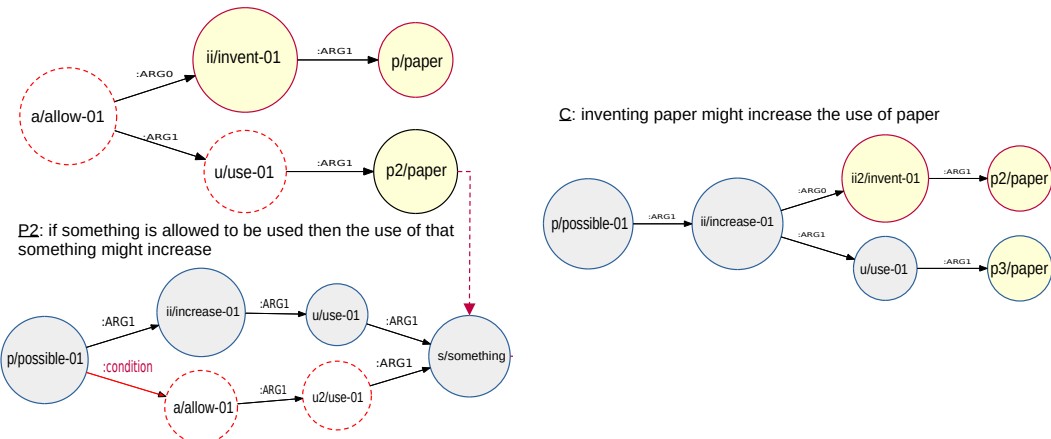

Figure 4: AMR conditional frame insertion (COND-FRAME).

**ARG-INS** (Figure 5): the conclusion is obtained by connecting an argument from one of the premises to a frame of the other.

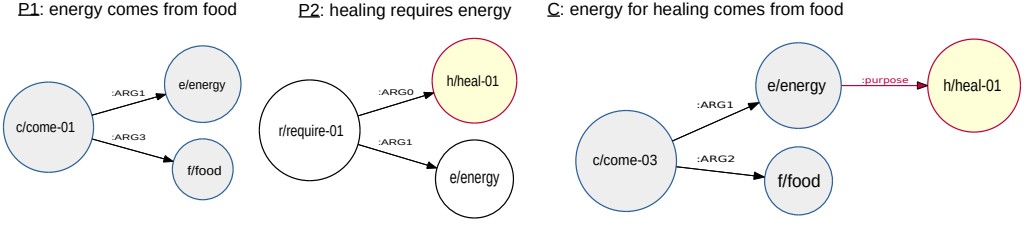

Figure 5: AMR argument insertion (ARG-INS).

**FRAME-CONJ**: the conclusion is obtained by using connectives to connect two premises.

**ARG/PRED-GEN** (Figure 6): a new *:domain* relation frame is created in the conclusion if both premise graphs differ by a single predicate/argument term.

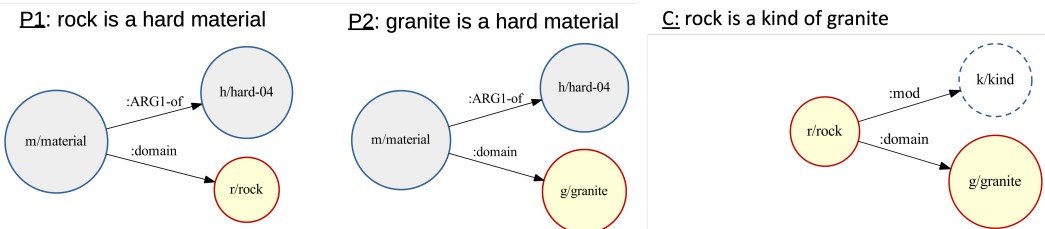

Figure 6: AMR argument generalisation (ARG-GEN).

**ARG-SUB-PROP** (Figure 7): one of the premises describes a "*is made of*" relationship between the entity in the other premise and its replacement.

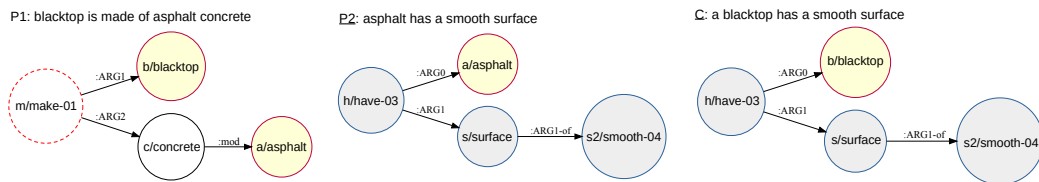

Figure 7: AMR argument substitution (property inheritance) (ARG-SUB-PROP).

**IFT**: the conclusion should be a conditional sentence. **EXAMPLE**: the conclusion should contain the keyword "example". **Unknown (UNK) category.** In this work, our annotation occupies 84% based on the EntailmentBank corpus. As for other unknown categories, we do not further specify them, as they either require knowledge outside of the scope of the premises or do not have a consistent symbolic transformation expression. An additional subtype called *premise copy* was included for the cases where the conclusion has the same graph as one of the premises.

## B EXPERIMENTAL DETAILS

### B.1 DATASET

In this work, we consider explanatory sentences as the testbed. The rationale for choosing them is that they are designed for formal, localised, symbolic semantic inference task in natural language form, which provides a semantically complex and yet controlled experimental setting, containing a both well-scoped and diverse set of target concepts and sentence structures, providing a semantically challenging yet sufficiently well-scoped scenario to evaluate the syntactic and semantic organisation of the space. Table 8 describes the statistical information of the corpus used in the experiment. For experiments: the EntailmentBank dataset is split into train 60%, valid 20%, and test 20% sets. For the explanation inference retrieval task, we follow the same experimental setup provided online[2].

Table 8: Statistics from explanations datasets. WorldTree is used in the Explanation Inference Retrieval task.

| Corpus | Num data. | Avg. length |
|---|---|---|
| WorldTree (Jansen et al., 2018a) | 11430 | 8.65 |
| EntailmentBank (Dalvi et al., 2021) | 5134 | 10.35 |

### B.2 T5 BOTTLENECK ARCHITECTURE

Figure 8 shows the architecture of the T5 bottleneck for learning latent sentence space. It includes two stages: sentence embedding and decoder connection. The sentence embedding aims to trans-

[2]https://github.com/ai-systems/hybrid_autoregressive_inference

form token embeddings into a sentence (single) embedding. Decoder connection aims to connect the encoder and decoder.

**Latent sentence space:** While designing the sentence bottleneck, we compare the four most frequently used mechanisms to transform token embeddings into sentence embeddings:

(1) Mean pooling: calculating the mean of each dimension on all token embeddings and feeding the resulting vector into a multi-layer perceptron to obtain the sentence embedding.

(2) multi-layer perceptron (MLP): applying an MLP to reduce the dimensionality of token embeddings, and the resulting embeddings are concatenated to form a single sentence embedding:

$$z = \text{concat}\Big[\text{MLP}_1(x_1); ...; \text{MLP}_T(x_T)\Big]$$

where $\text{MLP}_i(x_i)$ represents the $i$-th neural network for input representation of token $x_i$, $z$ is the latent sentence representation, and $T$ is the maximum token length for a sentence.

(3) multi-head attention: feeding each token embedding into the multi-head attention and considering the first output embedding as the sentence embedding (Montero et al., 2021):

$$z = \text{MultiHead}\left(XW^q, XW^k, XW^v\right)[0]$$

where $X = [x_1, ..., x_T]$ and $W^q$, $W^k$, and $W^v$ are the weights for learning $q$, $k$, $v$ embeddings in self-attention, respectively.

(4) Sentence T5: re-loading the pre-trained sentence T5 (S-T5, Ni et al. (2021)).

**Conditional generation:** Next, we consider four strategies to inject sentence embeddings into the decoder.

(1) Cross-attention input embedding (CA Input): reconstructing the token embeddings from a sentence representation and directly feeding them into the cross-attention layers of the decoder: $\hat{Y} = \text{MultiHead}\left(YW^q, \text{MLP}(z)W^k, \text{MLP}(z)W^v\right)$ where $\hat{Y}$ is the reconstruction of decoder input sequence $Y = [y_1, ..., y_K]$.

(2) Cross-attention KV embedding (CA KV): instead of reconstructing the token embeddings, it consists of directly learning the Key and Value (Hu et al., 2022; Li et al., 2020), which is formalised as $\hat{Y} = \text{MultiHead}\left(YW^q, \text{MLP}_k(z), \text{MLP}_v(z)\right)$, where $\text{MLP}_k$ and $\text{MLP}_v$ are neural layers for learning $k$ $v$ embeddings.

(3) Non-cross-attention input connection (NCA Input): reconstructing the token embeddings and element-wisely adding them with the input embeddings of the decoder (Fang et al., 2021).

(4) Non-cross-attention output connection (NCA Output): adding the reconstructed token embeddings to the output embedding of the decoder. The code is provided in the codebase[3].

Table 9: Comparison of different setups on test loss via cross-entropy (CA: cross-attention, NCA: non-cross-attention), bottom: comparison of different baselines on EntailmentBank testset.

| | | *Train: architecture* | | | |
|---|---|---|---|---|---|
| Decoder Connection | | CA Input | CA KV | NCA Input | NCA Output |
| | Pooling | 1.41 | 1.44 | 1.86 | 2.42 |
| Sentence | MLP | 1.71 | 1.94 | 2.09 | 2.62 |
| Embedding | MHA | 1.51 | 2.24 | 2.31 | 3.03 |
| | S-T5 | **1.24** | 1.42 | 1.81 | 2.22 |

### B.3 Implementation Details

**Hyper-parameters.** **1.** Size of Sentence Representation: in this work, we consider 768 as the size of the sentence embedding. Usually, the performance of the model improves as the size increases.

---

[3]https://anonymous.4open.science/r/Inference_type-5E07/

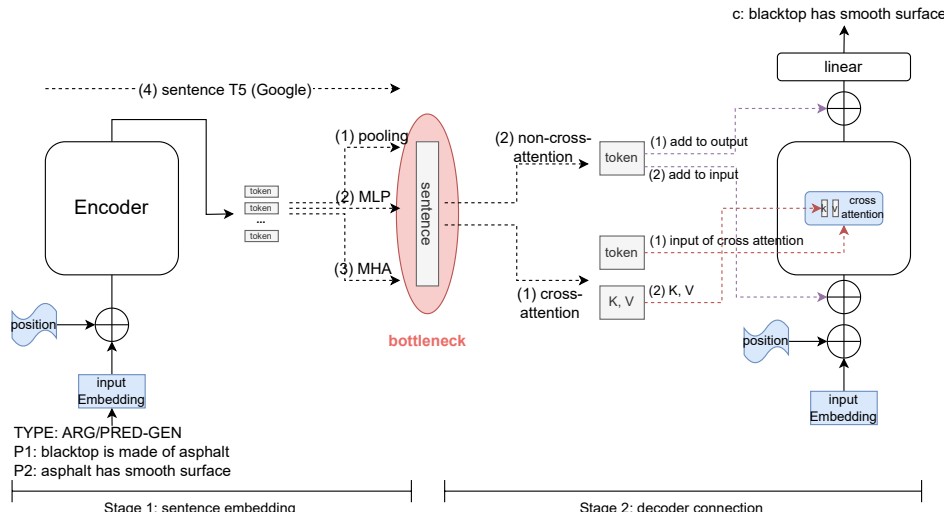

Figure 8: The architectural configuration of T5 bottleneck, it consists of two stages: sentence embedding and decoder connection.

**2.** Multi-head Attention (MHA): in the experiment, MHA consists of 8 layers, each layer containing 12 heads. The dimensions of Query, Key, and Value are 64 in each head. The dimension of token embedding is 768. Training hyperparameters are: **3.** For all models, the max epoch: 40, learning rate: 5e-5. During fine-tuning the T5 bottleneck, we first freeze the pre-trained parameters in the first epoch and fine-tune all parameters for the remaining epochs. **4.** All models are trained on a single A6000 GPU device.

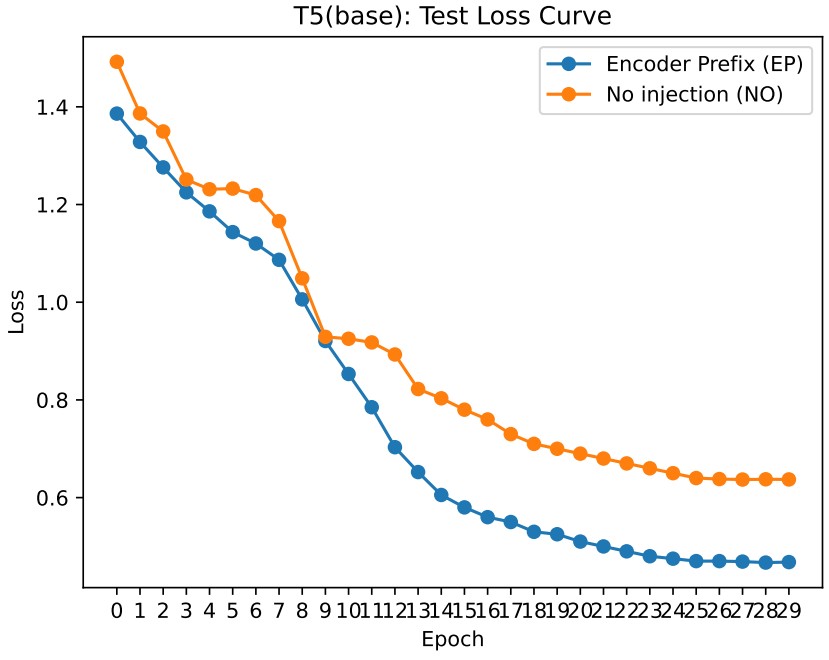

Figure 9: The test loss curve indicates that EP facilitates better convergence, indicating the supervision on inference types aligns the model's reasoning trajectory with target inference behaviours, improving conclusion prediction accuracy.

**Baselines.** In the experiment, we implement five LSTM-based autoencoders, including denoising AE (Vincent et al. (2008), DAE), $\beta$-VAE (Higgins et al., 2016), adversarial AE (Makhzani et al. (2015), AAE), label adversarial AE (Rubenstein et al. (2018), LAAE), and denoising adversarial autoencoder (Shen et al. (2020), DAAE). Their implementation relies on the open-source codebase available at the URL [4]. As for transformer-based VAEs, we implement Optimus (Li et al., 2020)[5] and Della (Hu et al., 2022)[6]. All baseline models undergo training and evaluation with the hyper-parameters provided by their respective sources. A latent dimension of 768 is specified to ensure a uniform and equitable comparative analysis.

**Metrics.** To evaluate the generated conclusions against the reference conclusions, we employ BLEU scores for 1- to 3-gram overlaps and report the average score. Additionally, to assess semantic similarity, we calculate the cosine similarity between the generated and reference conclusions by encoding both using the pretrained Sentence-T5 model[7] and computing the cosine similarity of their resulting embeddings.

## C  COMPLEMENTARY RESULTS

**Ablation studies.** We remove the inference types from the dataset and evaluate the T5 model performance using the same metrics. In this case, we can compare the model performance trained with or without that inference type. From Table 10, we can observe that the baselines (T5 small and base) achieve higher BLEU and BLEURT scores without the data with ARG-INS, COND-FRAME, and UNK inference type, respectively. This result indicates that the T5 cannot generalize well over those inference types. Also, removing the UNK inference type from data can achieve lower loss and PPL, which indicates that it has a negative impact on model training.

Table 10: Ablation study over inference type (No: no inference types are removed).

| Remove | T5 | BLEU | BLEURT | Cosine | Loss ↓ | PPL ↓ |
|---|---|---|---|---|---|---|
| FRAME-SUB | small | 0.50 | 0.19 | 0.95 | 0.95 | 2.58 |
| | base | 0.60 | 0.33 | 0.96 | 0.72 | 1.95 |
| ARG-INS | small | **0.54** | **0.27** | 0.95 | 0.82 | 2.22 |
| | base | **0.63** | **0.46** | 0.97 | 0.64 | 1.73 |
| FRAME-CONJ | small | 0.53 | 0.26 | 0.96 | 0.84 | 2.28 |
| | base | 0.60 | 0.35 | 0.96 | 0.65 | 1.76 |
| COND-FRAME | small | **0.55** | **0.25** | 0.96 | 0.88 | 2.39 |
| | base | **0.59** | **0.36** | 0.96 | 0.69 | 1.87 |
| UNK | small | **0.55** | **0.23** | 0.95 | 0.53 | 1.44 |
| | base | **0.62** | **0.40** | 0.96 | 0.58 | 1.57 |
| No | small | 0.54 | 0.22 | 0.96 | 0.69 | 2.22 |
| No | base | 0.57 | 0.33 | 0.96 | 0.61 | 1.65 |

**More controllable inference examples.** We provide more controlled examples based on both the Original T5 and T5 bottleneck in Table 11, 12, and 15. All examples reveal that the inference type can provide quasi-symbolic inference control to language models.

**Qualitative evaluation for LLM evaluators.** We conduct a qualitative evaluation through manual inspection. However, this assessment is not systematic or rigorously structured as we discussed in the Limitations section. Tables 13 and 14 present examples with discrepancies in scores between ChatGPT4o and GPT4o-mini, as well as a comparison of predictions between encoder prefix injection (EP) and the absence of inference-type injection (NO), respectively.

From both tables, we observe that ChatGPT4o tends to be more accurate than GPT4o-mini and that EP outperforms NO in generating correct predictions.

---

[4] https://github.com/shentianxiao/text-autoencoders
[5] https://github.com/ChunyuanLI/Optimus
[6] https://github.com/OpenVLG/DELLA
[7] https://huggingface.co/sentence-transformers/sentence-t5-base

Table 11: Controlled generation. original T5(base) (top) and T5 bottleneck (bottom).

**Quasi-symbolic NLI control**

P1: a pumpkin contains seeds
P2: fruit contains seeds

Original T5:
ARG-INS: a fruit in a pumpkin contains seeds
FRAME-CONJ: a pumpkin and fruit both contains seeds
FRAME-SUB: fruit is a kind of pumpkin

- - - - - - - - - - - - - - - - - - - - - - - - - - - - - - - - - - - - -

T5 bottleneck:
ARG-INS: fruit is a part of pumpkin that contains seeds
FRAME-CONJ: a fruit contains seeds
FRAME-SUB: a pumpkin is a kind of plant

Table 12: Controlled generation. original T5(base) (top) and T5 bottleneck (bottom).

**Quasi-symbolic NLI control**

P1: eating something has a negative impact on that something
P2: some animals eat cacti
ARG-INS: some animals have a negative impact on cacti by eating cacti
PRED-SUB: some animals may have a negative impact on cacti
FRAME-SUB: eating cacti has a negative impact on that cacti

- - - - - - - - - - - - - - - - - - - - - - - - - - - - - - - - - - - - -

ARG-INS: some animals have a negative impact on cacti by eating cacti
PRED-SUB: animals have a negative impact on cacti
FRAME-SUB: eating cacti has a negative impact on that cacti

Table 13: Qualitative evaluation for examples with discrepancies in scores between ChatGPT4o and GPT4o-mini (NO: no inference type injection, 0: invalid, 1: valid). We can observe that the ChatGPT4o tends to be more accurate than GPT4o-mini by human check.

| Premises | Prediction(NO) | Golden | ChatGPT4o | GPT4o-mini | Human Check |
|---|---|---|---|---|---|
| p1: the metal on the roof of a car is in contact with air

p2: air contains oxygen and water vapor | the car roof contains water vapor and oxygen | the metal on the roof of a car is in contact with oxygen and water vapor | 0 | 1 | 0 (invalid predicate "contains") |
| p1: friction occurs when the student is rubbing his hands together

p2: friction causes the temperature of an object to increase | rubbing your hands together causes the temperature of the object to increase | friction causes the temperature of student's hands to increase | 1 | 0 | 1 (replacing "friction" with "rubbing hands together") |
| p1: a caterpillar is a kind of insect

p2: metamorphosis is when an insect changes from an immature form to an adult form | metamorphosis is when a caterpillar changes from an immature form to an adult form | an example of metamorphosis is when a caterpillar changes from an immature form to an adult form | 1 | 0 | 1 (replacing "insect" with "metamorphosis") |
| p1: an increase in water has a positive impact on alligators
p2: a flood is caused by an increase in water | a flood has a positive impact on alligators | a flood has a positive impact on alligators | 1 | 0 | 1 (exact match) |
| p1: predators eat prey

p2: eating prey requires catching prey | predators catching prey requires catching prey | predators must catch prey to eat prey | 0 | 1 | 0 (fail to do substitution between "eating" and "catching") |
| p1: a leaf uses chlorophyll to produce carbohydrates
p2: carbohydrates are made of sugars | a leaf uses chlorophyll to produce sugars | a leaf uses chlorophyll to produce sugar | 0 | 1 | 1 (valid inference) |
| p1: salt is a kind of pure substance

p2: pepper is a kind of substance | salt and pepper are kinds of substances | salt and pepper are two substances | 1 | 0 | 1 (valid conjunction both "salt" and "pepper") |
| p1: some bacteria are good for humans by helping digestion

p2: digestion is when stomach acid breaks down food | some bacteria help digestion by breaking down food | some bacteria are good for humans by helping to break down food | 0 | 1 | 0 (without "good for human") |
| p1: a doorbell is a kind of electric device

p2: closing an electric circuit causes an electrical device to function | closing a doorbell causes the doorbell to function | an electric circuit causes a doorbell to function | 0 | 1 | 0 (incorrect replacing "electric circuit" with "doorbell") |
| p1: burning something is a kind of action

p2: if something occurs by performing an action, then that action is required for that something to occur | burning something is required for something to occur | if something occurs by burning something else, then burning that something else is required for that something to occur | 1 | 0 | 1 (valid inference by replacing "action" with "burning something") |
| p1: echolocation can be used for hunting prey

p2: bats must emit sound in order to echolocate | bats must emit sound in order to hunt prey | bats must emit sound in order to hunt their prey | 1 | 0 | 1 (valid inference by replacing "echolocate" with "hunting prey") |
| p1: different solids will have the same physical properties
p2: an mixture is formed by two or more substances combined together physically | one solid will form a mixture | different solids that are combined will become a mixture | 1 | 0 | 0 (incorrect "one solid") |

Table 14: Qualitative evaluation for prediction through EP and NO (NO: no inference type, EP: encoder prefix), we can observe that EP outperforms NO in generating correct predictions.

| Premises | Prediction(NO) | Prediction(EP) | Golden | ChatGPT4o | Human Check |
|---|---|---|---|---|---|
| p1: the metal on the roof of a car is in contact with air

p2: air contains oxygen and water vapor | the car roof contains water vapor and oxygen | the car roof is in contact with oxygen and water vapor | the metal on the roof of a car is in contact with oxygen and water vapor | NO:0, EP:1 | NO:0, EP:1 |
| p1: a beak is used for catching food by some birds
p2: eating food requires catching food | ads are used for eating by birds to catch food | a beak is used for eating by some birds | a beak is used for eating food by some birds | NO:0, EP:1 | NO:0, EP:1 |
| p1: predators must catch prey to eat prey
p2: a predator is a kind of animal | animals must catch and eat prey | animals must catch prey to eat prey | some animals must catch prey to eat | NO:0, EP:1 | NO:0, EP:1 |
| p1: an adaptation is a kind of change

p2: adapting is when a living thing responds to a change in an environment | an adaptation is something a living thing responds to a change in an environment | adaptation is when a living thing responds to a change in an environment | an adaptation is a kind of change in response to a change in an environment | NO:0, EP:1 | NO:0, EP:0 |
| p1: a doorbell is a kind of electric device

p2: closing an electric circuit causes an electrical device to function | closing a doorbell causes the doorbell to function | closing an electric circuit causes a doorbell to function | an electric circuit causes a doorbell to function | NO:0, EP:1 | NO:0, EP:1 |
| p1: green plants are made of plant cells
p2: green plants are a kind of producer | a producer is made of plant cells | producers are made of plant cells | producers are made of plant cells | NO:1, EP:0 | NO:1, EP:1 |
| p1: the iron nail has rusted


p2: rusting is when iron chemically reacts with water and oxygen | iron nails rusting is when the iron nails chemically react with water and oxygen to form iron nail | the iron nail has rusted | a chemical reaction has happened on the iron nail | NO:0, EP:1 | NO:0, EP:0 |
| p1: wood burns

p2: wood chips are made of wood | wood burns when introduced to wood | wood chips burn | wood chips burn | NO:0, EP:1 | NO:0, EP:1 |
| p1: some bacteria are good for humans by helping digestion
p2: digestion is when stomach acid breaks down food | some bacteria help digestion by breaking down food | some bacteria are good for humans by helping digestion by breaking down food | some bacteria are good for humans by helping to break down food | NO:0, EP:1 | NO:0, EP:1 |
| p1: a lung is a kind of organ for directly breathing in gasses
p2: smoke is a kind of gas | a lung is an organ directly for directly breathing in smoke | a lung is a kind of organ for directly breathing in smoke | lungs can be used for directly breathing in smoke | NO:0, EP:0 | NO:1, EP:1 |
| p1: the hot egg will decrease in temperature


p2: the cold water will increase in temperature | the hot egg and cold water will increase in temperature | the hot egg will decrease in temperature and the cold water will increase in temperature | the water will increase in temperature and the egg will decrease in temperature | NO:0, EP:1 | NO:0, EP:1 |
| p1: plant reproduction requires pollinating animals for pollination
p2: a bee can help on pollination by carrying pollen | plants reproduction requires bees that carry pollen | a bee can help plant reproduction by carrying pollen | a bee can help on pollination in plant reproduction by carry pollen | NO:0, EP:1 | NO:0, EP:1 |
| p1: a leaf uses chlorophyll to produce carbohydrates
p2: carbohydrates are made of sugars | a leaf uses chlorophyll to produce sugars | a leaf uses chlorophyll to produce sugar | a leaf uses chlorophyll to produce sugar | NO:1, EP:1 | NO:1, EP:1 |

Table 15: controllable NLI via inference type (Top: original T5, bottom: T5 bottleneck).

| Premises | Inference Type | T5 original |
|---|---|---|
| P1: a pumpkin contains seeds | ARG-INS | a fruit in a pumpkin contains seeds |
| P2: fruit contains seeds | FRAME-CONJ | a pumpkin and fruit both contain seeds |
| | IFT | if a pumpkin contains fruit then the fruit may contain seeds |
| | EXAMPLE | fruit is an example of pumpkins being sown |
| | ARG/PRED-GEN | a pumpkin is a kind of fruit |
| | ARG-SUB | fruit can contain pumpkin seeds |
| | UNK | a pumpkin can contain seeds |
| | FRAME-SUB | fruit is a kind of pumpkin |
| P1: sunlight is a kind of solar energy | ARG-INS | solar energy is a kind of resource for plants that uses water |
| P2: water and sunlight are resources for plants | FRAME-CONJ | water and sunlight are resources for plants and are kinds of solar energy |
| | UNK | the resources for plants include water and solar energy |
| | ARG-SUB | water and solar energy are resources for plants |
| P1: to move something can mean to transfer something | ARG-SUB | flowing can mean to transfer energy |
| P2: flowing is a kind of movement for energy | INF | if something flows, then that energy will flow |
| | FRAME-CONJ | moving can transfer energy and mean flowing |
| | ARG-INS | flowing can be a kind of transfer of energy to another entity |
| | ARG/PRED-GEN | transferring energy with flowing can be seen as transferring energy |
| P1: if a pot is exposed to a stove then that pot may become hot | COND-FRAME | the pot will become hot |
| P2: the pot is exposed to a stove | EXAMPLE | an example of hot pot is a pot exposed to a stove |
| P1: eating something has a negative impact on that something | FRAME-SUB | eating cacti has a negative impact on that cacti |
| P2: some animals eat cacti | PRED-SUB | some animals may have a negative impact on cacti |
| | ARG-INS | some animals have a negative impact on cacti by eating cacti |
| | EXAMPLE | cooking cacti is an example of a negative impact on a cactus |
| | INF | if a cactus has a negative impact on an animal, that cactus could be devoured |
| P1: seeing requires light | ARG-SUB | reading requires light |
| P2: reading requires seeing | ARG-INS | light is a kind of requirement for reading |
| | INF | if light is moving then reading may be taken |
| | EXAMPLE | a light bulb will be used for reading |
| | UNK | light will help you read |

| Premises | Inference Type | T5 bottleneck |
|---|---|---|
| P1: a pumpkin contains seeds | ARG-INS | fruit is part of a pumpkin that contains seeds |
| P2: fruit contains seeds | FRAME-CONJ | a fruit contains seeds |
| | FRAME-SUB | a pumpkin is a kind of plant |
| P1: sunlight is a kind of solar energy | ARG-INS | water is a kind of resource that is used by plants for growth |
| P2: water and sunlight are resources for plants | FRAME-CONJ | plants and water are resources that require water and energy |
| | ARG-SUB | plants use water and sunlight to produce energy |
| P1: to move something can mean to transfer something | ARG-SUB | flowing can mean to transfer energy |
| P2: flowing is a kind of movement for energy | INF | if something flows, then that energy will flow |
| | FRAME-CONJ | moving can transfer energy and mean flowing |
| | ARG-INS | flowing can be a kind of transfer of something |
| | ARG/PRED-GEN | transferring energy with flowing can be seen as transferring energy |
| P1: if a pot is exposed to a stove then that pot may become hot | COND-FRAME | the pot may become hot |
| P2: the pot is exposed to a stove | ARG/PRED-GEN | the pot may be a source of heat |
| P1: eating something has a negative impact on that something | FRAME-SUB | eating cacti has a negative impact on that cacti |
| P2: some animals eat cacti | PRED-SUB | animals have a negative impact on cacti |
| | ARG-INS | some animals have a negative impact on cacti by eating cacti |
| P1: seeing requires light | ARG-SUB | reading requires light |
| P2: reading requires seeing | FRAME-CONJ | reading and feeling can both be used |
| | INF | if something is visible then that something will be seen |

**Algorithm 1** Annotation procedure

1: Find premise $P_x$ most similar to the conclusion $C$, $P_{\bar{x}}$ being the other premise.
2: $G_{x,\bar{x},C}$ ← AMR graph of $P_x, P_{\bar{x}}, C$, respectively.
3: # - - - - - - - - - - - - - - - - - - common ARG-SUB, PRED-SUB - - - - - - - - - - - - - - - - - - - - - -
4: **if** $G_x = G_c$ or $G_{\bar{x}} = G_c$ **then**
5:     $type = PREM\text{-}COPY$ # Comment: no reasoning happen.
6: **else if** $P_x$ and $C$ differ by one word $w$ **then** # Comment: common ARG(PRED)-SUB.
7:     **if** $w$ is a verb **then**
8:       $type = PRED\text{-}SUB$
9:     **else**
10:       $type = ARG\text{-}SUB$
11:     **end if**
12: **else**
13: # - - - - - - - - - - - - - - - - - COND-FRAME, FRAME-SUB, ARG-SUB-PROP - - - - - - - - - - - - - -
14:     Get AMR graphs $G_1, G_2, G_c$ for $P_1, P_2$ and $C$ respectively. $P_x \to G_x$.
15:     **if** $\exists$ :ARG*$(x,a) \in C$ and $a \in P_{\bar{x}}$ **then**
16:       **if** $\exists$ :condition$(root(G_x), root(G_{\bar{x}}))$ **then**
17:       # Comment: see Figure 4, two root nodes are connected by :condition edge
18:         $type = COND\text{-}FRAME$
19:       **else if** $root(a)$ is a noun **then**
20:         **if** $root(G_{\bar{x}}) = $ "make-01" and $\exists$ :ARG*$(root(G_{\bar{x}})$, a) **then**
21:         # Comment: "make" as a trigger to classify ARG-SUB and property inheritance.
22:           $type = ARG\text{-}SUB\text{-}PROP$
23:         **else**
24:           $type = ARG\text{-}SUB$ # ARG-SUB that was not caught by the simpler rule on line 10, due to Px differing from C by more than a single word
25:         **end if**
26:       **else**
27:         $type = FRAME\text{-}SUB$
28:       **end if**
29: # - - - - - - - - - - - - - - - - - - - Further-specification and Conjunction - - - - - - - - - - - - - - - - - - - - - -
30:     **else if** $G_x \subset G_c$ and $G_{\bar{x}} \subset G_C$ **then**
31:       $type = FRAME\text{-}CONJ$
32:     **else if** $\exists x, y$ :domain$(root(G_x), x)$ and :domain$(root(G_{\bar{x}}), y)$ and :op*("and", x) $\in G_c$ and :op*("and", y) $\in G_c$ **then** # Comment: using connectives 'and' to connect two premises
33:       $type = FRAME\text{-}CONJ$
34:     **else if** $G_x \subset G_c$ **then**
35:       $d \leftarrow G_c - G_x$
36:       **if** $root(d)$ is a noun **then**
37:         $type = ARG\text{-}INS$ # Comment: inserting an argument.
38:       **else**
39:         $type = FRAME\text{-}INS$ # Comment: inserting a phase (also annotated as ARG-INS).
40:       **end if**
41: # - - - - - - - - - - - - - - - - - - - ARG/PRED-GEN and Others - - - - - - - - - - - - - - - - - - - - - - - -
42:     **else if** $\exists$ :domain$(root(G_c), y)$ and $(root(G_c) \in G_x$ and $y \in G_{\bar{x}})$ or $(root(G_c) \in G_{\bar{x}}$ and $y \in G_x)$ **then**
43:       $type = ARG/PRED\text{-}GEN$
44:     **else**
45:       $type = UNK$
46:     **end if**
47: **end if**

Table 16: Empirically designed prompt for automatically evaluating the controllability in Section 5.2.

---

**Prompts for automatic evaluation**

**Consistency:**
You are a scoring expert in natural language reasoning. Given two premises and a conclusion, your goal is to evaluate whether the conclusion violates the premises. During your inference process, please only consider the information from the premises.
you can directly give your score (0 or 1) based on the following criteria:
0: the conclusion violates the premises.
1: the conclusion doesn't violate the premises.

The output format is just the score. You don't need to analyse the reasoning process.

- - - - - - - - - - - - - - - - - - - - - - - - - - - - - - - - - - - - - - - - - - - - - - -

**Alignment:**
You are a scoring expert. Given two premises, a conclusion, and an inference type, your goal is to evaluate whether the (premises, conclusion) pair is aligned with the inference type.

The following is the description of 10 inference types:
1. ARG-SUB: the conclusion is obtained by replacing one argument with another argument.
2. PRED-SUB: the conclusion is obtained by replacing one verb with another verb.
3. FRAME-SUB: the conclusion is obtained by replacing a frame of one of the premises with one from the other premise.
4. COND-FRAM: the conclusion is obtained according to the conditional premise with keyword "if".
5. ARG-INS: the conclusion is obtained by connecting an argument from one of the premises to a frame of the other.
6. FRAME-CONJ: the conclusion is obtained by using connectives to connect two premises.
7. ARG/PRED-GEN: a new ":domain" relation frame is created in the conclusion if both premise graphs differ by a single predicate/argument term.
8. ARG-SUB-PROP: one of the premises describes a "is made of" relationship between the entity in the other premise and its replacement.
9. IFT: the conclusion should be a conditional sentence.
10. EXAMPLE: the conclusion should contain the keyword "example".

When evaluating, some premises might not be able to deduce more than one conclusions. You can ignore those cases.

Finally, you can directly give your score (0 or 1) based on the following criteria:
0: the (premises, conclusion) pair is not aligned with the inference type.
1: the (premises, conclusion) pair is aligned with the inference type.

The output format is just the score. You don't need to analyse the reasoning process.

