# OpenReview forum: "Guiding Explanatory Inference through Inference Types"
_ICLR.cc/2026/Conference — Submitted to ICLR 2026_

### Official Review · Reviewer_qrnk · 2025-10-27

**Soundness:** 3
**Presentation:** 2
**Contribution:** 3
**Rating:** 6
**Confidence:** 3

**Summary:**

This paper investigates how to realize localized and quasi-symbolic inference behaviors in a distributed representation space. Specifically, the authors focus on the Explanation-based NLI task, which involves deriving a conclusion from two explanations (premises).

The paper's core contribution is the introduction of inference types, a set of systematically defined quasi-symbolic reasoning behaviors. Using Abstract Meaning Representation (AMR) as a formalization framework, the authors annotated 5134 samples from EntailmentBank, defining 10 symbolic transformation types.
Theoretically, the paper proposes a quasi-symbolic NLI framework and formalizes it based on the Neural Tangent Kernel (NTK) theory. The framework hypothesizes that different inference types can be encoded as functionally separate subspaces within the model's parameter space through gradient descent.

**Strengths:**

1.  This paper aims to address the lack of localized and quasi-symbolic inference behavior in NLI models, which is crucial for fields like explainability, controllability, and mechanistic interpretability.
2.  The use of AMR as an intermediate interface to define inference types is an good choice. This provides a solid linguistic and formal semantic foundation for the 10 inference types, rather than just ad-hoc labels.
3.  The paper not only proposes a framework but also attempts to use NTK theory to explain its underlying mechanism.

**Weaknesses:**

The framework relies on AMR to define inference types, and the annotations are performed manually. In a practical inference setting, how would the system determine which inference type should be applied to a new, unseen pair of premises? The paper seems to assume that the target inference type $\pi$ is always given. This is reasonable for a control task, but for the standard NLI prediction task, this seems to be a limitation. In a standard inference scenario where $\pi$ is unknown, how do you envision this framework working? Would it require a separate classifier to predict $\pi$ first? Or do you believe its primary application is in controllable text generation?

**Questions:**

Please refer the weaknesses section :-)

---

> ### Author Response · Authors · 2025-11-18
> **Response to Reviewer qrnk**
>
> Dear Reviewer qrnk,
>
> Thank you for the effort and time spent on reviewing the contribution. We address the main point raised below:
>
> **[Weakness1]** The current setup assumes a known inference type, as our primary goal is to study how symbolic reasoning behaviours can be explicitly represented and controlled within neural models from the perspective of latent space. This controlled setting allows us to analyse *“functional separation and interpretability”*, rather than to replace standard NLI inference pipelines.
>
> One potential direction is to investigate the controlled inference through VAE or Diffusion architecture, where the inference behaviours can guide the decoding stage, thus improving interpretability and controllability.

---

### Official Review · Reviewer_Vw6H · 2025-10-30

**Soundness:** 3
**Presentation:** 3
**Contribution:** 2
**Rating:** 4
**Confidence:** 3

**Summary:**

This paper investigates quasi symbolic reasoning within Transformer models on explanation-based nlp tasks. The task involves deriving a conclusion from two explanatory premises. The authors first introduce a formalization of this reasoning behavior by defining ten "inference types" grounded in abstract meaning representation.

The core technical contribution is a framework for guiding a T5 model using these explicit inference types. The authors provide a theoretical justification using neural tangent kernel theory, positing that different inference types induce distinct, separable function spaces within the model's parametric space. Empirically, they demonstrate this by feeding the inference type label as a prefix to the model's encoder. This approach improves generative accuracy on the NLI task and also enables localized control. The model can be guided by changing the prefix at inference time to produce structurally different conclusions from the same premises. Their analysis also shows that this method leads to disentanglement, with inference types forming more separable clusters in the latent and parametric spaces.

**Strengths:**

The paper's primary strength lies in its fusion of formal linguistic structures with standard neural architectures to achieve controllable reasoning. The introduction of AMR grounded "inference types" is a well structured formalism for characterizing the quasi symbolic steps in explanation based NLI.

The authors propose a method and a theoretical justification using neural tangent kernels, positing that these inference types map to distinct functional subspaces. This claim is validated through multiple experiments. The demonstration of localized control, where varying the inference type prefix systematically alters the generated conclusion from identical premises, is interesting.  It is supported by quantitative analyses showing increased separability in both the latent feature space and the parametric (gradient) space, which strengthens the theoretical hypothesis.

**Weaknesses:**

While the work is well-motivated, its technical execution and claims have several limitations. The theoretical justification using NTK is tenuous. NTK theory is most precise for infinite-width networks operating in a "lazy" training regime where parameters do not move far from initialization. This paper, however, deals with a fine-tuned, fixed-width model that is explicitly intended to learn new functional representations (feature learning), a dynamic that often deviates significantly from the lazy regime. The paper provides empirical evidence of gradient orthogonality, but it does not bridge the conceptual gap between the idealized NTK theory and the practical, non-lazy dynamics of fine-tuning T5.

The claims of "disentanglement" and "quasi-symbolic control" also are somewhat overstated. The model is learning prefix-conditioned generation, which is a form of control, but the evidence for true latent space separation is weak. The potential instability of this control can be observed in the qualitative examples. For instance, the "ARG-SUB" guided model generates "the blacktop is made of smooth surface," which is a semantic and grammatical error. This suggests the model is not executing a symbolic rule (substituting an entity) but is performing a soft, pattern-based substitution (replacing the text "asphalt concrete" with "smooth surface") that fails to respect the underlying semantic roles.

Finally, the use of AMR is a missed opportunity. AMR is used to *define* the inference type labels, but this structural information is never provided to the model. The model only sees a low-bandwidth string prefix. This means the model learns a single representation for "ARG-SUB," even though this category may contain many structurally distinct operations at the AMR graph level. A more rigorous test of quasi-symbolic grounding would involve conditioning the model on richer, graph-based structural information derived from AMR.

**Questions:**

1. Can you further justify the application of NTK theory? The fine-tuning of T5 seems to operate in a feature learning regime, which deviates from the lazy training assumptions inherent to NTK. How does the theory meaningfully apply in this practical, non-lazy context?

2. In the qualitative examples, the ARG-SUB operation produces "the blacktop is made of smooth surface," which is semantically incorrect. Does this error indicate that the model is learning a shallow textual substitution pattern associated with the prefix, rather than a robust, semantically-grounded symbolic rule?

3. The AMR graphs provide a rich structural basis for the inference types, yet the model is only conditioned on a single, coarse label like "ARG-SUB". Have you explored conditioning the model on a more direct structural representation of the AMR transformation itself, rather than this many-to-one textual label?

4. How do you expect this framework to generalize to entirely new inference types not present in the training data? Does the current approach learn the rules themselves, or does it primarily learn to associate ten specific labels with ten learned output behaviors?

---

> ### Author Response · Authors · 2025-11-18
> **Response to Reviewer Vw6H**
>
> Dear Reviewer Vw6H,
>
> Thanks for your effort and insightful reviews. We respond to your main points below:
>
> **[Weakness1 & Question1]** We agree that standard NTK theory is formally derived for infinite-width networks trained in the lazy regime, whereas fine-tuned T5 operates in a feature-learning regime. In our work, the NTK formulation is not used as a literal training assumption, but as a conceptual and analytical framework to formalize the idea that inference-type supervision induces functionally separated subspaces in parameter space. This abstraction allows us to connect quasi-symbolic reasoning to well-defined geometric properties (kernel similarity, gradient alignment) observed in practical networks.
>
> We fully acknowledge that the fine-tuned T5 operates in a feature-learning regime that deviates from the strict NTK limit. However, recent studies have shown that NTK-derived intuitions can empirically extend to finite-width models, where kernel-like behaviour still correlates with functional modularity and disentanglement (ref1). Our empirical results (Sec. 5.4, Fig. 3) demonstrate such correspondence through measurable gradient orthogonality and latent clustering.
>
> We will clarify this scope in the revision, explicitly framing NTK as a conceptual approximation for analyzing quasi-symbolic functional subspaces rather than a restrictive training assumption.
>
> Ref1: “Task Arithmetic in the Tangent Space: Improved Editing of Pre-Trained Models”  https://arxiv.org/pdf/2305.12827
>
> **[Weakness2]** Qualitative examples such as “the blacktop is made of smooth surface” illustrate the soft and approximate nature of the model’s reasoning (integrating some formal structure with content-based features in the context of material inferences). However, our goal is not to achieve flawless linguistic generation, but to examine *whether neural models can encode and control distinct reasoning behaviours in the latent space in a quasi-symbolic manner*, thereby enabling systematic manipulation and interpretability of the underlying inference processes. As shown in Table 6, the >60% consistency and alignment scores indicate that most generated conclusions remain logically coherent with the premises, supporting that the control is systematic rather than random.
>
> The term *“disentanglement”* in our work specifically refers to *“functional separation of inference behaviours in parameter space”*. Section 5.4 provides quantitative evidence for this through gradient orthogonality and latent-space clustering, showing that different inference types occupy distinguishable functional subspaces. This supports the notion of *“quasi-symbolic control”*: a form of soft, rule-based modulation over reasoning trajectories. This stands in contrast to prior studies (e.g., ref2 and ref3), which argue that the learned function space becomes increasingly entangled or that model inference largely reflects memorisation rather than rule-based reasoning.
>
> As the initial step in this direction, we focus on the coarse-grained inference task. Future studies can focus on the finer-grained inference structures (e.g., syllogisms), to strengthen semantic coherence and stability.
>
> Ref2: “Questioning Representational Optimism in Deep Learning: The Fractured Entangled Representation Hypothesis” https://arxiv.org/pdf/2505.11581
>
> Ref3: “Do PhD-level LLMs Truly Grasp Elementary Addition? Probing Rule Learning vs. Memorization in Large Language Models” https://arxiv.org/abs/2504.05262v1
>
> **[Weakness3]** As stated in Section 3 (L146–148), AMR in our framework is used to systematically define and annotate inference behaviours, not as a representational component of the model itself. Our central objective is to examine whether a neural model can function as a soft, rule-based reasoner when given only minimal symbolic supervision, without explicit structural conditioning. This design isolates the effect of inference-type control from that of complex graph inputs, allowing us to probe whether quasi-symbolic behaviour can emerge naturally in standard neural architectures.
>
> We agree that incorporating explicit AMR graph information and encoding it via an independent graph encoder (like the current multi-modal model) could enrich the model’s grounding and provide a more rigorous test of structural reasoning, which can be considered as a promising direction to explore more coherent logical and structural inference control.

---

> > ### Author Response · Authors · 2025-11-18
> > **Continue**
> >
> > **[Question2]** This behaviour reflects a known limitation of quasi-symbolic reasoning in current neural models: the model correctly performs the intended structural transformation (argument substitution) but lacks full semantic grounding to ensure naturalness of the resulting phrase. Our goal is precisely to make such phenomena measurable and interpretable, i.e., to separate the control of inference behaviour from lexical content, so that structural reasoning patterns can be analysed independently of surface fluency.
> >
> > The quantitative evaluations in Sec. 5.2 (Tab. 6) show that most generated conclusions remain logically consistent with the premises, supporting that the prefix signals induce systematic inference control rather than random text substitution. We have added clarifications to emphasise that quasi-symbolic behaviour represents an intermediate regime between surface patterning and robust symbolic reasoning, and future work will integrate semantic validators (e.g., AMR-based reconstruction) to further strengthen grounding.
> >
> > **[Question3]** We intentionally conditioned the model on a coarse inference-type label (e.g., ARG-SUB) to isolate and analyse how symbolic supervision alone shapes representational subspaces, without conflating the effect with potential noise or parsing variability from full AMR inputs. This minimal conditioning allowed us to directly test whether distinct inference behaviours could emerge as separable latent functions under lightweight symbolic control.
> >
> > With that said, our framework is representation-agnostic by design: the inference-type label can be replaced or augmented with structured AMR encodings describing the specific graph transformation (e.g., subgraph substitution templates). We will clarify this future direction in the revision, noting that structured conditioning offers a natural extension of our quasi-symbolic formulation. The exploration of conditioning on explicit AMR edit graphs to capture fine-grained transformation paths while preserving interpretability should be explored in the future.
> >
> > **[Question4]** Key contribution: The current study uses ten inference-type categories as a controlled testbed to probe whether neural NLI models can internalise and separate distinct reasoning behaviours. While the model is conditioned on discrete labels during training, it does not simply memorise label-output pairs. Instead, it learns functional mappings between premise representations and reasoning transformations, as evidenced by gradient orthogonality and latent clustering (Sec. 5.4).
> >
> > Conceptually, this framework is extensible beyond the observed labels. Because each inference type corresponds to an operation (grounded on universal linguistic structures) the model’s latent representation captures parametric regions associated with reasoning dynamics rather than fixed categories. This structure supports compositional interpolation: new inference types can be expressed as combinations or variants of existing transformations. We plan to explore this in future work by evaluating zero-shot and compositional generalization to unseen symbolic operations.

---

### Official Review · Reviewer_sW2W · 2025-10-31

**Soundness:** 3
**Presentation:** 3
**Contribution:** 3
**Rating:** 2
**Confidence:** 3

**Summary:**

The paper studies explanation-based NLI by making reasoning controllable and interpretable. To make this happen, contributions are (1) linguistic formalization, where they annotate dataset with inference types that describe how to transition from premise to conclusion (annotated dataset is planned for public release) and (2) quasi-symbolic framework where they teach transformers (e.g., T5) to govern subspace formation in their parametric space. The method is shown to improve accuracy and controllability.

**Strengths:**

1. Using inference type to teach transformers is elegant and novel. The paper reads well and the core idea is quite interesting.

2. The framework has a strong theoretical support (using AMR and NTK).

3. The detailed annotation as well as the annotated dataset (from EntailmentBank) itself will help reproducibility and also the research community in general.

4. Empirical analysis are detailed and show improved performance and interpretability.

**Weaknesses:**

1. The method currently uses two premises which is quite limited in real-world, so I was wondering if how the approach can handle multi-hop reasoning situations with more premises involved.

2. Most of the core experiments and analyses were based on specific model T5, so the validation is kind of skewed IMO (e.g., separated subspaces is only validated using T5 architecture).

3. Annotating by humans is usually expensive, therefore expanding to include other domains is quite challenging.

**Questions:**

1. As mentioned in weaknesses, can authors show how to handle multi-hop reasoning in their setting?

2. Authors mentioned math reasoning as future work, so I'd be really interested to see how their approach (e.g., inference types) would be adapted for math?

3. How could we expand the framework to more (known or unknown number) of premises?

---

> ### Author Response · Authors · 2025-11-18
> **Response to Reviewer sW2W**
>
> Dear Reviewer sW2W,
>
> Thanks for your feedback and insightful reviews. We respond to your main points below:
>
> **[Weakness1]** Our primary motivation and contribution lie in analysing the geometric structure of the latent space, with a focus on enabling quasi-symbolic NLI control and interpretability, an area that remains largely under-explored, rather than pursuing a direct real-world application.
>
> As an initial step, we focus on the two-premise setting, as this provides a *“minimal and controlled environment”* for characterising quasi-symbolic inference types and their representational separation. Importantly, the two-premise/one-conclusion configuration corresponds to *a typical syllogistic-style inference structure*, which has been widely adopted in both formal logic and explanation-based NLI datasets (e.g., EntailmentBank). This design choice allows us to systematically analyse discrete reasoning behaviours without confounding interactions among multiple hops.
>
> Nevertheless, the framework extends naturally to multi-hop reasoning. In fact, our entailment tree retrieval experiment (Table 7) already demonstrates its capability to handle reasoning chains up to four hops by iteratively composing inference-type transformations. Each local two-premise operation can be viewed as a symbolic edge in a larger entailment graph, enabling compositional reasoning over multiple steps.
>
> Our contribution may further inspire the neuro-symbolic community by demonstrating the feasibility of learning human-interpretable inference behaviours within neural models, thereby enabling more interpretable and controllable NLI systems design that can systematically memorise, retrieve, and infer knowledge, rather than a black-box model. For example, a growing body of research has demonstrated that in many instances these models often rely on memorisation rather than generalisation and that rule-based control mechanisms fail to be fully enforced (ref1).
>
> Ref1: “Do PhD-level LLMs Truly Grasp Elementary Addition? Probing Rule Learning vs. Memorization in Large Language Models” https://arxiv.org/abs/2504.05262v1
>
> **[Weakness2]** We primarily focus on the T5 architecture because its encoder-decoder separation naturally aligns with our theoretical formulation, where the encoder instantiates the quasi-symbolic reasoning function and the decoder performs natural language realisation. This design enables controlled evaluation of subspace formation and inference-type separation.
>
> Nonetheless, our framework is not tied to T5, as we claimed in section 4.2. We additionally validate key findings across *decoder-only* models (GPT-2, Qwen2.5, LLaMA3.2) and *encoder-bottleneck-decoder* architectures (Optimus) in Tables 2-4, observing consistent gains when inference-type supervision is introduced. Moreover, our in-context learning experiments (Table 4) show that the same inference-type conditioning improves reasoning behaviour in large chat models, confirming cross-architecture generalisability.
>
> **[Weakness3]** We appreciate the reviewer’s insightful point regarding annotation cost and scalability. Indeed, manual annotation of inference types requires expert effort; however, we intentionally designed the scheme to be *systematic (rule-book codified), linguistically grounded, and transferable across domains*. The ten inference-type categories are defined via AMR-based symbolic transformations, which make the annotation process highly structured and suitable for semi-automatic extension using pattern matching or LLM-based labeling.
>
> Moreover, our current annotation (5,134 pairs) serves as a foundational benchmark for studying quasi-symbolic reasoning, analogous to early manually curated resources (e.g., AMR, EntailmentBank). Once the inference-type taxonomy is established, future datasets can be expanded automatically by leveraging LLM-assisted annotation and weak supervision guided by our formal templates.

---

> > ### Comment · Reviewer_sW2W · 2025-11-25
> > **Increase score**
> >
> > I appreciate author responses. After reading them as well as other reviews, I would like to increase my score. Despite this, I remain interested to know how to extend this work to math reasoning (i.e., second question).

---

> > > ### Author Response · Authors · 2025-12-01
> > > **Response to Reviewer sW2W**
> > >
> > > Dear Reviewer sW2W,
> > >
> > > Thank you for your engagement and for raising your score. Regarding your question: the math reasoning task refers specifically to the symbolic math task (ref1), for example, applying differentiation rules such as sin(x) -> cos(x) or log(k) -> 1/k, .etc. This task provides a systematic framework for constructing a rule-based symbolic reasoning corpus.
> > >
> > > Our findings show that inference can be separated and controlled within the latent space during gradient optimisation. Building on this insight, we adopt a more controlled experimental setup using symbolic mathematical expressions to further explore rule-based learning and generalisation, and knowledge memorisation, retrieval and representation in transformer language models.
> > >
> > > ref1: A Symbolic Framework for Evaluating Mathematical Reasoning and Generalisation with Transformers https://aclanthology.org/2024.naacl-long.84.pdf

---

### Official Review · Reviewer_eP1G · 2025-11-01

**Soundness:** 3
**Presentation:** 3
**Contribution:** 3
**Rating:** 6
**Confidence:** 3

**Summary:**

This paper presents a quasi-symbolic framework for natural language inference. The method involves three stages: (1) the premises are encoded into neural representations, (2) those representations are transformed conditional on an inference type, and (3) the representations are transformed into a conclusion. This enables guidance during training and interpretability during inference. By controlling the inference type, the generated conclusions can be controlled

Strengths:
- The family of inference types seems reasonably broad and varied to me, and the detailed information on the neuro symbolic architecture was interesting and well presented.
- The neuro-symbolic approach seems fascinating, and its intuitive that structuring of inferences would help interpretability.
- The method seems to work, with natural language inference results improving, though see my question below

Weaknesses:
- Generally, the neurosymbolic approach seems like it falls prey to the "bitter lesson". I think this research is really cool and full of deep ideas, but I can help but think that in practice if we want a better neural NLI system we would want to context engineer chat models with these inference types. I don't think that is a reason to reject this paper, as I still think that the neuro-symbolic approach could bear fruit someday, but it does prevent me from strongly advocating for this line of work.
- The connections to NTK were not particularly helpful for me, and I felt like my intuitions about how this training would operate wasn't helped by this. Maybe work on this connection to explain how its a useful lens for the work?
- The qualitative analysis was not that surprising or illuminating for me. It makes sense that these inference types can be separated in space, and the structure didn't look striking, kind of seems like what would have to happen if the training was successful and the inference types were used.


Just to double check, the performance in table 2 for the NO row is still on a model that was fine-tuned for this task correct? I just want to make sure you aren't comparing a pretrained model against your method for training a model.

Overall, I think this type of work is cool and this paper is worth accepting.

**Strengths:**

See summary

**Weaknesses:**

See summary

**Questions:**

See summary

---

> ### Author Response · Authors · 2025-11-18
> **Response to Reviewer eP1G**
>
> Dear Reviewer eP1G,
>
> We thank the reviewer for the positive assessment of our contribution and for recognizing the novelty and clarity of our neuro-symbolic framework. We address the main concerns below.
>
> **[weakness1]** We agree that large, context-engineered LLMs can often achieve strong empirical NLI performance. However, our work is not positioned as a direct competitor to scaling-based approaches, but rather as *a complementary investigation into the representational mechanisms that underlie (quasi-)symbolic reasoning in neural models.*
>
> The quasi-symbolic framework provides a theoretical and diagnostic lens for understanding how inference-type supervision shapes latent reasoning dynamics, offering a pathway toward interpretable and controllable NLI behaviour. Moreover, our in-context learning experiments (Table 4) show that the same inference-type conditioning can improve reasoning accuracy in large models, indicating that our approach can inform and enhance context engineering. In this sense, our contribution is to characterise how symbolic reasoning signals can be integrated within neural architectures as an initial step.
> Since our results reveal the possibility of rule-based learning and control within neural networks, future work will focus on how we can systematically design or optimise the latent space to induce such prior knowledge in the latent space to assist the memorising, retrieving and inferring.
>
> **[weakness2]** NTK is a theoretical framework for interpreting training dynamics, model behaviour, and latent geometry. This directly matches our objective: we want inference types to correspond to separable functional subspaces so that we can control them locally, targeting neuro-symbolic NLI behaviour (a soft rule-based approach).
>
> NTK framework tells us that:
>
> (1) Generalisation and inference behaviours during training are controlled by the geometry of gradients.
>
> (2) By measuring gradient cosine similarity across inference types, we are directly measuring how much optimisation for one type affects the other.
>
> (3) When we observe low cross-type similarity, this means gradient descent is effectively carving out separate functional subspaces for each inference type: training on ARG-SUB mainly changes the ARG-SUB behaviour, training on FRAME-CONJUNCTION mainly changes the FRAME-CONJUNCTION behaviour, etc.
>
> Thus, the NTK connection is intended to give a concrete picture of optimisation: training with inference-type labels is equivalent to kernel regression where each type defines its own pattern of sharing and interference, and our diagnostics (gradient similarities, latent separation) are direct probes of that optimisation geometry.
>
> **[weakness3]** The qualitative analysis complements our quantitative findings (Fig. 3 left) by showing that the separation is *“functionally meaningful”*: the clusters correspond not just to label memorisation but to distinct compositional behaviours (e.g., frame conjunction vs. argument substitution) that can be *“systematically manipulated”* to generate different valid conclusions (Tab. 5). This indicates that symbolic reasoning behaviours can be localized and controllably activated within the latent space, rather than being a by-product of training.
>
> Furthermore, we stress that the emergence of interpretable, geometrically separable subspaces is not guaranteed by successful training alone, many multi-task or label-conditioned models fail to produce such structure without explicit regularization or disentanglement constraints. Our results show that lightweight symbolic supervision alone (via inference-type prefixes) suffices to induce this disentanglement, providing empirical evidence for the feasibility of rule-based representation learning in vanilla neural NLI models.
>
> Moreover, this study also provides some interesting findings that provide a more nuanced understanding from  previous work (e.g., ref1), suggesting that higher task performance does not necessarily correspond to more structured or disentangled internal representations. Instead, the learned function space may become increasingly entangled as a result of SGD optimisation.
>
> Ref1: “Questioning Representational Optimism in Deep Learning: The Fractured Entangled Representation Hypothesis” https://arxiv.org/pdf/2505.11581
>
> **[Answer to the Question]** Yes, that is correct.

---

### Author Response · Authors · 2025-12-01
**Summary**

Dear ICLR program committee,

We summarise the main concerns from reviewers and our response (revised in the paper with blue colour) below.

**1. The motivation and implementation of NTK is unclear:** NTK theory offers a principled interpretation of training dynamics, knowledge memorisation, retrieval, and representation, showing that generalisation and inference behaviours are shaped by the geometric structure of gradients throughout training. The relevance of NTK-based interpretations is supported by recent studies demonstrating that NTK approximations can remain informative even when the model undergoes optimisation.

**2. AMR is not a practical input representation, misaligned with the annotation:** In this study, we exclude AMR representations from the input to minimise the influence of explicit structural linguistic biases on the internal geometry learned by the vanilla Transformer NLI model. Instead, only the inference-type labels are provided as input to explore the "rule-based learning and control". Incorporating formal AMR structures into neural representations requires systematic investigation and likely architectural modifications. As this work represents an initial step toward that goal, we view the integration of AMR-based structural information as an important direction for future research.

**3. Limitation in the broader domain:** Our work concentrates on deeper mechanistic interpretability and controllability from the standpoint of rule-based learning and inference control, an important yet underexplored direction within the neuro-symbolic literature. We argue that this perspective can offer new insights into knowledge representation, retrieval, and reasoning by examining how these processes are organised within the model’s latent geometry.

We would appreciate it if these points could be considered in the final decision. Thanks for your hard work.

---

### Meta-Review · Area_Chair_JRLQ · 2026-01-07

**Summary:**

This paper studies controllable, quasi-symbolic reasoning for explanation-based NLI. It introduces a taxonomy of 10 AMR-grounded “inference types” and uses them to condition a Transformer (mainly T5) to (i) improve generative NLI accuracy and (ii) enable controllable conclusion generation by changing the inference-type prefix. The paper further argues, via an NTK-inspired lens and empirical probes (e.g., gradient similarity / clustering), that inference-type supervision induces more separable functional subspaces, supporting interpretability and localized control.

Pros
1. Clear and interesting neuro-symbolic framing: inference-type conditioning provides an intuitive control knob for reasoning behavior.
2. The AMR-grounded inference-type formalization and curated annotations/dataset are potentially valuable to the community.
3. Empirical results suggest improvements in accuracy and controllability, and analyses (latent/gradient separation) support the “functional separation” hypothesis.

Cons
1. The NTK justification remains tenuous for fine-tuned, finite-width, feature-learning regimes. The rebuttal reframes NTK as a conceptual/diagnostic lens, but the gap to formal NTK assumptions is still not fully bridged, so the theoretical contribution feels weaker than implied.
2. Evidence for “disentanglement” / quasi-symbolic rule execution is mixed: some qualitative examples look like brittle prefix-conditioned pattern substitution with semantic/grammatical errors, raising concerns about stability and grounding.
3. The use of AMR is limited to defining labels; the model only sees a coarse prefix rather than richer structure. This can make the quasi-symbolic grounding feel low-bandwidth and many-to-one.
4. Practicality/generalization questions remain: scaling beyond two premises / multi-hop settings, transfer to other domains/models, and how to select inference type when it is not given (standard NLI setting). Annotation cost and expanding the scheme beyond the current dataset are also concerns.

The idea and dataset are interesting and the controllability results are promising, but a stronger revision would (i) sharpen/limit theoretical claims, (ii) strengthen empirical evidence for stable, semantically grounded control, and/or (iii) explore richer structural conditioning and more realistic settings (multi-premise, inference-type prediction).

**Reviewer Concerns:**

1. Why use NTK / what it contributes: Authors clarify NTK is a conceptual/analytic lens (gradient geometry, kernel similarity) rather than a strict lazy-training assumption, and they commit to tightening wording in the revision.
2. Two-premise limitation / multi-hop: Authors argue the 2-premise setup is a controlled testbed and point to an iterative multi-hop/tree-style experiment as evidence the framework can compose over multiple steps.
3. Architecture dependence (mostly T5): Authors claim additional validation on decoder-only and other architectures (plus in-context experiments) showing the inference-type signal helps beyond T5.
4. Annotation scalability concern (partially): Authors outline a path to semi-automatic/LLM-assisted labeling and stress the rule-book structure of the taxonomy.
5. “Bitter lesson” / relevance vs chat model prompting: Authors position the goal as interpretability/control and suggest the inference-type concept can also inform prompting/in-context learning, not only training smaller models.

**Reviewer Scores:**

N/A

---

### Decision · Program_Chairs · 2026-01-26

Reject